# Physical measures of physical functioning as prognostic factors in predicting outcomes for neck and thoracic pain: Protocol for a systematic review

Rabea Begum[ID]*, Alison Rushton[ID], Alaa El Chamaa[ID], David Walton, Paul Parikh

School of Physical Therapy, Western University, London, Ontario, Canada

* mbegum6@uwo.ca

## Abstract

### Background

Spinal pain is prevalent and burdensome worldwide. A large proportion of patients with neck and thoracic pain experience chronic symptoms, which can significantly impact their physical functioning. Therefore, it is important to understand factors predicting outcome to inform effective examination and treatment. Knowledge of physical measures of physical functioning as prognostic factors can enhance patient-centered care and aid decision-making. The evidence regarding physical outcome measures as prognostic factors for neck and thoracic pain is unclear. The objective of this study is to summarize the evidence for physical outcome measures of physical functioning as prognostic factors in predicting outcomes in people with neck and thoracic pain.

### Methods and analysis

This systematic review follows Cochrane guidelines and aligns with the Preferred Reporting Items for Systematic Review and Meta-Analysis Protocols (PRISMA-P). Included studies will be prospective longitudinal cohort studies in which physical measures of physical functioning are explored as prognostic factors for adults with neck and thoracic pain. A comprehensive search will be performed in key databases (MEDLINE, EMBASE, CINAHL, Scopus, and Web of Science) and the grey literature, with hand searches of key journals, and the reference lists of included studies. Two reviewers will independently perform study selection, data extraction, risk of bias assessment (QUIPS, Quality in Prognostic Studies tool), and quality assessment (Grading of Recommendations Assessment, Development, and Evaluation).

### Implications

This systematic review will identify physical measures of physical functioning prognostic factors for neck and thoracic pain populations. Findings will inform researchers about gaps in

**Data Availability Statement:** The research data will be made publicly available when the study is completed and published.

**Funding:** The author(s) received no specific funding for this work.

**Competing interests:** The author have declared that no competing interests exist.

existing evidence, and clinicians about factors to aid their clinical decisions and to enhance the overall quality of care for individuals with neck and thoracic pain.

## Introduction

Spinal pain is a significant public health issue with neck pain which is a prevalent musculoskeletal disorder in adults [1] ranking as the fourth most common cause of years lived with disability [2–4]. According to the latest Global Burden of Disease study, neck pain affected 203 million people in 2020, with peak ages between 45–74 years old, and it will affect 270 million people by 2050 [5]. Thoracic pain is less common than neck and low back pain and its 1-year prevalence is about 15% compared with 32% and 43% for neck and low back pain, respectively [6]. Evidence suggests a significant number of people experience neck pain (70%) [7] and thoracic pain (4–77%) at least once during their lifetime [8]. Research investigating thoracic spine is scarce [9]. While many cases of acute neck pain will resolve, current estimates indicate that the majority (50–85%) will recur or become chronic within 5 years [10, 11]. This burden has a substantial impact on physical functioning [12], health-related quality of life (HRQoL) [12–15], activity limitations, and disability [13, 16] and thus imposes a significant personal and socioeconomic burden [2]. Similarly, thoracic pain can be disabling, and it can cause significant disability [17] and similar burdens on the individual's life [18] though the burden of thoracic pain has not been adequately established [8]. Compared with the neck and low back, the thoracic spine has received less attention, probably owing to the public health burden of thoracic pain being presumed to be less [19]. Thoracic and cervical impairments often co-exist [20], making the thoracic region a critical component of examination, classification, diagnosis, and management of neck pain, as per clinical practice guidelines [21, 22]. Understanding both conditions together is crucial due to the shared anatomical and biomechanical relationship between the neck and thoracic spine. Dysfunction in cervical spine might affect thoracic spine and vice versa. Dysfunction in thoracic spine is considered a potential contributor which necessitates the need to explore the contribution of thoracic region toward dysfunction in cervical spine [23].

The IMMPACT (Initiative on Methods, Measurement, and Pain Assessment in Clinical Trials) provides a standardized framework for assessing outcomes in pain management and emphasizes that physical functioning is an important independent outcome domain which is also included in the core outcome set (COS) for a meaningful evaluation of pain [24]. For purposes of this protocol, physical functioning is conceptualized as a multidimensional construct that integrates body structures and function, activity and participation capacity, influenced by both personal and social factors, per the international Classification of Functioning, Disability, and Health (ICF) [25]. Measuring limitation of physical functioning is important because it can affect a person's quality of life, disability, and health care expenses [26].

Measures of physical functioning evaluate diverse aspects of a participant's life, including the ability to carry out such daily activities as household chores, walking, work, travel, and self-care, as well as strength and endurance [27]. Physical functioning can be assessed through various methods such as patient-reported outcome measures (PROMs) or physical measures (observable, quantifiable) of impairment, performance, or activity [24, 28–30]. PROMs are standardized tools used to evaluate various aspects of perceived or anticipated physical functioning from the patient's perspective [31]. Impairments are characterized by dysfunction or structural abnormality of a specific body function or structure [32]. Physical measurement of

impairments assess structural or functional limitation, such as limited range of motion (ROM) (measured by a goniometer or inclinometer), and diminished grip strength (measured by a dynamometer) [33]. Physical measures of performance measure can be used to quantify physical function in a standardized environment (e.g., hospital, clinical setting). For these measures, a participant is asked to perform a specific task and performance on the task is evaluating to identify limitations in physical functioning, such as the Timed Up and Go (TUG) test [24], and Functional Capacity Evaluation (FCE) [34] which consists of a series of tests (lifting, pushing, pulling, and other tasks). Physical measures can also be used to provide estimates of physical function in real environments typically captured through technologically advanced measurement devices like accelerometer which is widely used to measure sedentary time, physical activity, and sleep-related behaviors [24, 35]. Physical measures of impairment, performance, and physical measures of activity in a real environment provide different insights into the functioning of the body, and when interpreted together can provide integrated insights into physical function beyond what any one measure can provide in isolation [24]. Many healthcare providers globally, including physiotherapists, extensively utilize physical measures in clinical practice. MacDermid and colleagues [36] have performed an international survey investigating outcome measures used by clinicians for evaluation of neck pain. Physical measures were frequently used, with 44% clinicians using ROM, 44% using segmental joint mobility, and more than 56% using neck muscle strength, along with PROMs. Clinicians also routinely assess joint mobility, tissue extensibility and function using ROM [37], mobility, endurance, and strength measures when evaluating people with thoracic pain [38].

There are several systematic reviews [39–44] that have synthesized physical prognostic factors for individuals with neck pain. They investigated a range of physical prognostic factors, including ROM [39–44], joint position error (JPE) [39–41], altered muscle recruitment [40, 41]. However, cervical motor dysfunctions, such as reduced ROM, disturbed cervical kinesthesia, muscle strength/endurance and altered cervical muscle activity have limited predictive value for long-term (6-month, 1 year, 2 year or 3 year) outcomes [39, 40]. These systematic reviews exclusively focus on acute (<6 weeks) whiplash injuries; however, sub-acute (6 to 12 weeks) [22] and chronic (>12 weeks) symptoms [22] were not explored for other types of neck pain (e.g. radicular pain). The majority included studies only in the English language. Only one systematic review included other languages (German, French and Dutch) along with English [42]. Language restrictions in systematic reviews should be avoided due to language bias [45], particularly with recent innovations in online translation. In addition, the overall risk of bias assessment score of these systematic reviews is high according to AMSTAR-2, (A Measurement Tool to Assess Systematic Reviews) [46]. Key measures for high risk of bias assessments include: risk of bias not assessed for included studies [41, 44], grey literature not included [40–44] and level of evidence not assessed [40, 41]. Only one systematic review [39] used a standard approach (GRADE) for level of evidence. No review included evaluation of performance-based, and activity measures in the natural environment. Most of the systematic reviews investigated prognostic factors for acute (< 6 weeks) whiplash condition. Previous systematic reviews focused psychophysical measures, for example cold pain threshold [40, 41, 43, 44] and pressure pain threshold [43, 44] while investigating physical prognostic factors for neck pain. This emphasis highlights a gap of physical functioning. Therefore, there is a considerable knowledge and methodological gap in the existing literature regarding physical functioning measures for predicting outcomes in neck pain.

Regarding thoracic pain, research investigating prognostic factors is lacking. There are only two systematic reviews [8, 47] that have synthesized different factors for thoracic pain. The first systematic review by Briggs and colleagues [8] studied associated risk factors in the general population and they concluded that the majority of studies included were cross-sectional

in design which limiting inferences about prognosis for thoracic pain and they recommend including prospective cohort studies for providing evidence more robust evidence for prognostic factors. The second systematic review [47] did not find any studies investigating prognostic factors for thoracic pain [47]. To date there are no systematic reviews that have investigated physical measures of physical functioning as prognostic factors for thoracic pain population and clinicians face challenges in seeking information about prognostic factors for thoracic pain.

Owing to the methodological issues detailed above, and insufficient inclusion of physical measures in systematic reviews of prognostic factors, a gap in existing evidence exists for neck and thoracic pain population. It is therefore important to systematically determine prognostic factors predicting outcomes following neck and thoracic pain.

## Objective

To summarize the evidence for physical measures of physical functioning as prognostic factors for predicting outcomes in individuals with neck and thoracic pain.

## Methods

### Design

This systematic review protocol is reported according to the Preferred Reporting Items for Systematic Review and Meta-Analysis Protocols 2015 statement [48] (S1 File), the Cochrane Back Review Group guidelines [49] and the Cochrane Handbook [50]. This protocol is registered in the International Prospective Register of Systematic Reviews (PROSPERO) to improve transparency, accountability, and prevent duplication [51], registration ID: CRD42024574473.

### Eligibility criteria

The PICOS (Population, Intervention/Exposure, Comparator, Outcome and Study design) framework is used to develop the inclusion criteria [52]. The Comparator component is not applicable due to the nature of this review.

**Population (P).** Individuals aged 18 and above with neck and thoracic pain for any duration (acute or chronic) will be included. For the purpose of this review, the anatomical boundaries of neck and thoracic pain will be used as reported in the primary studies.

**Exposure (E).** Studies will be included if they use any of the physical measures of physical functioning, including impairment-based physical measures (e.g., strength, ROM) [29], performance-based physical measures (e.g., TUG), and activities in the natural environment (e.g., accelerometry) [24].

**Outcome (O).** No limitation will be placed on the types of outcomes evaluated. Including all outcomes provides a more comprehensive understanding of the potential prognostic factors. The scoping search has identified the limited number of studies available on this topic, and restricting the outcomes might result in excluding valuable data. The limited number of studies available emphasized the necessity for a wider range of outcomes in our review. Any time point of outcome assessment will be included.

**Study design (S).** Only prospective longitudinal cohort studies will be included as they are the preferred design for establishing a clear temporal sequence between exposure and outcome [53].

No limitation on language and publication year will be applied. For non-English studies, open-source software, specifically ChatGPT (Chat Generative Pre-Trained Transformer) will be used to translate the texts, and bilingual individuals will verify the translation and confirm

the accuracy. These individuals will have a background of health sciences, allowing them to identify and correct any errors, particularly with medical terminology.

## Exclusion criteria

Studies where pain is a result of a space-occupying lesion (e.g., tumors, cysts), any serious neurological disorders (e.g., spinal cord injury, myelopathy), disc infection, inflammatory disorders (e.g., ankylosing spondylitis), fibromyalgia, any metabolic bone diseases (e.g., osteoporosis), deformities (e.g., congenital scoliosis), spinal dislocation or fracture and other studies related to the non-neuromusculoskeletal origin of pain will be excluded from this review.

## Information sources

A comprehensive systematic search will be performed in electronic databases from inception to 31st July 2024 using databases: MEDLINE (Ovid), EMBASE (Ovid), CINAHL, Scopus, and Web of Science. Theses and dissertations will be searched using ProQuest Dissertations and Theses Global and conference proceedings/abstracts in Web of Science and Scopus. Manual (or hand) search will be performed in key journals: Spine, European Spine Journal and The Spine Journal. The reference lists of the studies will also be thoroughly screened.

## Search strategy

The search strategy was informed by scoping searches. The search strategy has been developed in MEDLINE (Ovid) in collaboration with a library information specialist, Western University. The MEDLINE search comprised medical subject heading (MESH) terms and text words based on PICOS format, for example populations (individuals with neck and thoracic pain), exposures (physical measures of physical functioning) and study design (prospective longitudinal cohort study). Firstly, the alternative MESH terms and keywords added with OR (Boolean operator) for each concept and after that the three concepts combined with AND (Boolean operator) following Cochrane guideline for searching studies [54]. This search will be adapted in other databases with modifying syntax. The search terms mentioned in the previous systematic reviews [55, 56] and scoping search were used for text words. The MEDLINE (Ovid) search is presented in the S2 File.

## Data management

For data management, Covidence, a web-based software platform for systematic reviews will be used [57]. Duplication will be automatically identified and removed by the Covidence software.

## Study selection process

Two independent reviewers (RB and AE) will perform the eligibility assessment, initial screening of titles and abstracts, and full-text screening according to eligibility criteria. Excluded studies which do not meet the criteria, will be reported with all reasons for exclusion. The two reviewers will meet and discuss in case of any differences in assessment. Any differences between the two reviewers will be resolved by a third reviewer (P.P) who will act as a mediator. Agreement between reviewers will be evaluated with Cohen's kappa coefficient where values $\leq 0$ as indicating no agreement and 0.01–0.20 as none to slight, 0.21–0.40 as fair, 0.41–0.60 as moderate, 0.61–0.80 as substantial, and 0.81–1.00 as almost perfect agreement [58]. The process of study selection will be reported in the PRISMA flow diagram [59] (Fig 1).

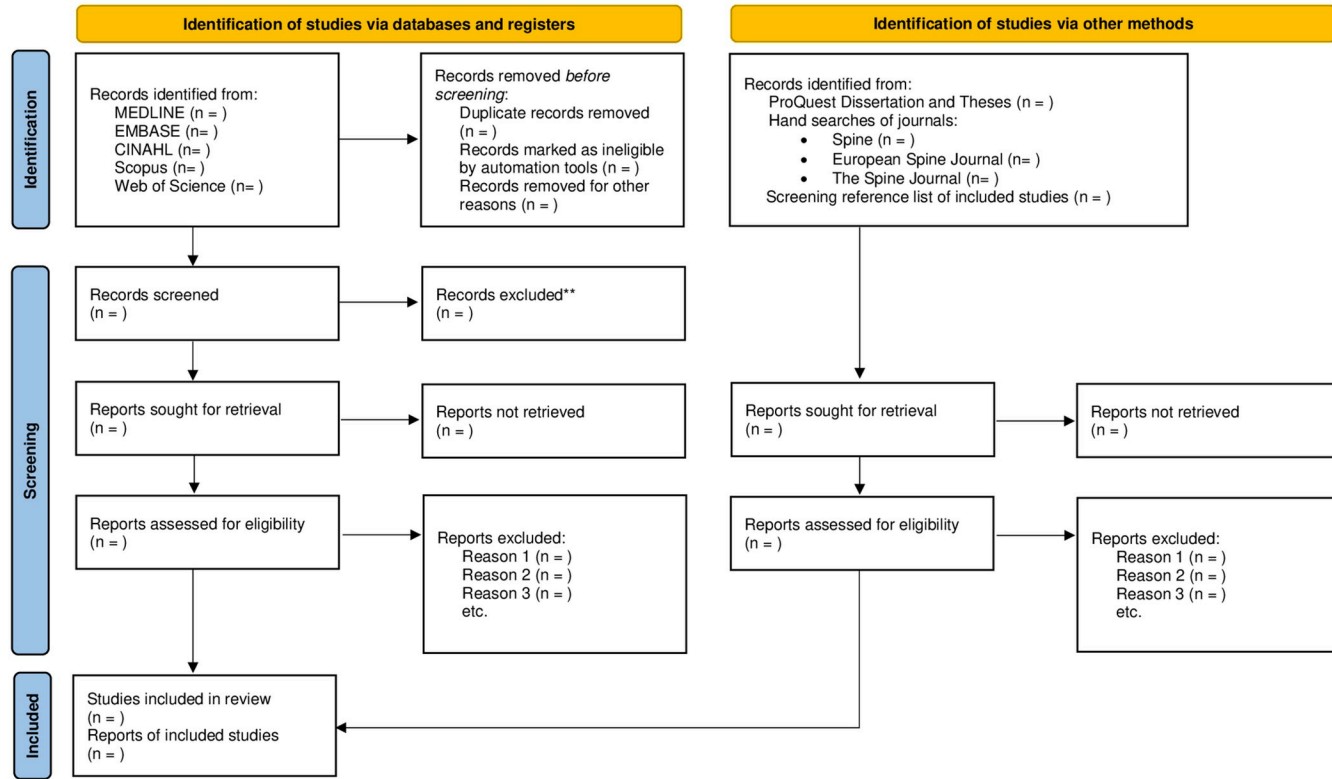

**Fig 1. PRISMA 2020 flow diagram.** This diagram reflects the study selection process.

## Data extraction process

The CHecklist for critical Appraisal and data extraction for systematic Reviews of prediction Modelling Studies (CHARMS) will be used to extract data. CHARMS provides a checklist and background explanation for extracting data for prognostic studies [60]. Two reviewers will extract data independently from selected studies. If disagreement, a third reviewer (P.P) will mediate. The corresponding author of the included studies will be contacted in cases of any data missing or unclear data. A reminder email will be sent after 3 weeks. If there are no responses within a further 3 weeks, we will extract available data.

## Data items

Data will be extracted from the included studies based on the CHARMS checklist [60] (Table 1).

## Outcomes and prioritization

Outcomes will be categorized as impairment-based, performance-based, and activities in a natural environment. The outcome data will be extracted into short-term (< 3 months), medium-term (3 months to 12 months), and long-term (>12 months) [61].

**Table 1. Summary of data to be extracted from included studies.**

| Characteristics of study | Title, Authors, year of publication, study design, country of study, objective(s) of the study, source of funding |
|---|---|
| Characteristics of participants | Age, gender, history of neck and thoracic pain, characteristics of neck and thoracic pain (e.g., duration, distribution of pain), sample size |
| Physical measures of physical functioning candidate prognostic factors | Name, type (e.g., impairment-based measure), equipment required |
| Outcome measures | Name, type (e.g., PROMs or physical measures), equipment required, time points of assessment of outcome (follow up duration) |
| Results | Analysis method, type of statistical measure, main findings |

## Risk of bias (RoB) of individual studies

Two reviewers will use QUIPS to assess RoB according to Cochrane Prognosis Methods Group recommendations [62]. The QUIPS tool is a standard quality assessment method for included studies in a systematic review and it contains 6 domains. 1) Study participation will assess the representativeness of the study sample and the potential for selection bias. Here we will also consider the information provided on the baseline characteristics of study participants to evaluate the risk of selection bias. 2) Study attrition will evaluate the impact of participant loss on results and the completeness of follow-up. We will investigate the reasons for loss to follow-up and aim to maintain an attrition rate of ≤20%. 3) Prognostic factor measurement will examine the reliability and validity of how prognostic factors are measured. 4) Outcome measurement will focus on the clarity and standardization of outcome definitions and assessments. 5) Study confounding will determine if potential confounding factors are appropriately controlled to isolate effects. 6) Statistical analysis, and reporting will evaluate the appropriateness and transparency of the statistical methods used in the analysis [63]. The different domains contain 3 to 7 prompting items to be rated (yes, partial, no, unsure) and each domain will be rated on a three-graded scale as low, moderate, or high RoB [64]. A low risk of bias will indicate that the findings are reliable, with the true result expected to be close to the estimate. Moderate risk will suggest the true result may vary, requiring careful interpretation. High risk will raise concerns that the true result could significantly differ from the estimate, affecting validity [63, 64]. Before starting the review, two independent reviewers (RB, AE) will undergo calibration exercise using a subset of studies to ensure consistent application of the QUIPS tool. They will provide rationales for any discrepancies and engage in structured discussions based on pre-defined criteria, focusing on study design, participant selection, prognostic factors, and outcome assessment. This process will ensure a consistent, reliable risk of bias assessment and support well-informed decisions. Any disagreements will be discussed between two independent reviewers and if a consensus is not reached, a third reviewer (PP) will mediate. The level of agreement will be evaluated by using Cohen's kappa coefficient [58].

## Data synthesis

The synthesis of data for each prognostic factor will be conducted using either a meta-analysis (quantitative synthesis), such as pooling data through meta-analysis, or a narrative approach (qualitative synthesis), by providing a narrative summary of the findings.

**Assessment of heterogeneity.** The decision to perform a quantitative synthesis will be based on the assessment of heterogeneity (e.g., clinical heterogeneity, methodological heterogeneity, and statistical heterogeneity) in included studies. The variability in the participants, outcomes, outcome measures, follow-up duration, and prognostic factors will be measured by

clinical heterogeneity, the variability in RoB will be assessed as methodological heterogeneity, and statistical heterogeneity will be the differences in the effects of prognostic factors and outcomes due to clinical and methodological differences [65, 66]. The $I^2$ statistic and Cochrane Q test. The $Chi^2$ statistic will be used to test and quantify statistical heterogeneity. The $Chi^2$ test will assess the differences in results across studies, and a low p-value ($p<0.05$) will suggest a significant heterogeneity. The $I^2$ statistic will calculate the proportion of total variation due to heterogeneity than due to chance. The $I^2$ value ranges from 0–100% and it can be interpreted as 0%-40%: might not be important, 30%-60%: moderate heterogeneity, 50%-90%: substantial heterogeneity, and 75%-100%: considerable heterogeneity [67].

**Meta-analysis or quantitative synthesis.** If studies are sufficiently homogenous in terms of physical measures of physical functioning prognostic factors or outcomes, a meta-analysis will be performed [67, 68]. The summary measures will depend on the nature of the outcomes. For dichotomous outcomes, risk ratios (RR) or odds ratios (OR) will be calculated. For continuous outcomes, mean differences (MD) or standardized mean differences [69] will be used. A random effect model with 95% confidence intervals (CI) will be performed to estimate the effect across studies. A forest plot will be used to display an overall estimate of the effect [70, 71]. The initial scoping search suggested substantial clinical heterogeneity across existing studies, indicating considerable differences in prognostic factors, or outcomes. This variability limits the feasibility of conducting a meaningful subgroup analysis. The limited number of studies identified in the initial search will also constrain sensitivity analysis in the meta-analysis plan, particularly regarding exploring the influence of our judgment on ROB [72]. The influence of high RoB studies will be carefully considered, and they may be given less weight in the synthesis. Their potential impact on the results will be acknowledged, and they will be considered when assessing the overall quality of evidence in the GRADE evaluation. All statistical analyses will be conducted using R statistical software [73].

**Narrative analysis or qualitative synthesis.** In the situation of a meta-analysis not being possible (due to methodological and clinical heterogeneity), a systematic narrative or qualitative synthesis will be conducted to summarize findings [68, 74]. We will follow three major steps except the first one (developing a theory of how the intervention works, why and for whom) due to nature of this review. Step 1: the preliminary synthesis of the findings will be conducted by summarizing and organizing of included studies according to populations categories, outcomes, potential physical measures of physical functioning candidate prognostic factors. Step 2: the strength of association of different outcomes based on the point of outcome assessment will be explored to find out the relationships in the data within and between studies. Step 3: to explain differences between and within studies, variability of outcomes, populations, and physical measures will be investigated. We will use the GRADE approach to provide a systematic method to assess the certainty in evidence and a transparent documentation of body of evidence [75]. The narrative synthesis will present text and tables to organize and discuss the data and methodological aspects of the included studies.

*Dealing with missing data*. We will handle missing data in this stage by attempting to calculate missing statistics manually, where feasible. For example, if a study reports standard errors (SE) without reporting standard deviations (SD), we will estimate the missing standard deviations using the relationship between standard errors and standard deviations ($SD = SE + \sqrt{n}$, where $n$ is the sample size). If manual calculation is not possible, we will follow appropriate statistical methods outlined in the Cochrane Handbook, such as imputation method [76]. In a narrative synthesis, the impact of missing data will be addressed by discussing the impact of missing data on the overall findings which will ensure transparency and reliability of the findings.

## Meta-bias(es)

If any protocols are found during our searches, reporting bias will be evaluated by examining the consistency of study protocols and published results.

## Confidence in cumulative evidence

Two independent reviewers will use the GRADE (Grading of Recommendations Assessment, Development, and Evaluation) approach [77] to assess the quality of evidence of included studies. It will be categorized into five domains: risk of bias, inconsistency, indirectness, imprecision, and publication bias [78]. The quality evidence can be upgraded or downgraded [79]. The QUIPS tool will be for risk of bias and rated as having no serious limitation, serious limitation and very serious limitation [80]. The evidence will be downgraded if many studies have high risk of bias. Then, to assess the consistency of different prognostic factors across included studies, $I^2$ value will be used to quantify heterogeneity. It will be downgraded in case of significant unexplained heterogeneity. After that, the evidence will ensure the physical functioning measures and outcomes they predict are directly relevant and it will be downgraded if studies use different populations, outcomes that are not directly applicable. Additionally, the evidence will be considered imprecise if confidence intervals are wide or sample sizes are small, and it will be downgraded if there is high uncertainty around effect estimates. Finally, funnel plots will be used to detect publication bias if possible. It will downgrade the evidence if there is evidence of publication or selective reporting bias [79]. The GRADE approach will specify evidence into four levels of certainty consist of 1) high: very confident that true effect lies close to the estimate of the effect, 2) moderate: moderately confident in the effect estimate, 3) low: confidence in the effect estimate is limited, and 4) very low: very limited confidence in the effect estimate [78]. In three situations, the confidence might increase (rating up)-large effect, dose-response gradient or plausible confounding. The large effect will provide significant differences in outcomes between groups exposed to different prognostic factors, indicating that the observed association is unlikely to be due to bias or confounding factors. A dose-response relationship can occur when changes in exposure, such as muscle strength links to changes in the likelihood of an outcome. The domain plausible confounding will be described the effect of the confounding factor on the estimate [79]. For example, if several high-quality included studies will show a specific physical measure (e.g., ROM) is a strong predictor of neck pain outcomes, with a large effect size (relative risk or RR >2 or <0.5), it could be justified as rating the evidence up. If the included studies show that greater ROM is associated with better neck or thoracic outcomes consistently across multiple studies, it will be considered as the rating up. Finally, if well-designed studies will account for confounding outcome variables (e.g., other health conditions or comorbidities) and find a strong association between physical measures and neck or thoracic pain outcomes and the observed effect will not be due to these confounding factors, in that case, it will suggest the effect is robust and rate the evidence up. Two reviewers will try to solve any disagreements through discussion. A third reviewer will resolve if a consensus cannot be reached. Detailed information and certainty of the evidence will be reported in the 'Summary of findings' tables [79].

## Ethics and dissemination

Ethics approval is not required due to the nature of the study design and there will be no direct contact with patients. The results of this review will be disseminated through publication in a peer-reviewed journal.

## Patient and public involvement

This protocol has been discussed with the spinal pain research Patient Partner Advisory Group in the School of Physical Therapy at Western University. Key feedback included 1) the proposed review protocol was very clear, 2) the advisory group expressed their lived experiences with neck and thoracic pain, and they highlighted the importance of using physical measures alongside patient-reported outcome measures and 3) lastly, their feedback also guided the selection of a broader range of outcomes. Specifically, their feedback emphasized the need to include not only clinical outcomes (e.g., pain, disability) but also those that reflect day-to-day functioning and quality of life, which are highly relevant to individuals living with neck and thoracic pain. The feedback ensured that it would be understandable and usable for both researchers and patients. The Patient Partner Advisory Group did not suggest any modifications to the physical measures which supports the review's approach.

## Discussion

Neck and thoracic spinal pain are prevalent musculoskeletal disorders that can have significant personal and social burdens. People experiencing spinal pain report some consequences on physical functioning, for example- reduced daily activities that can result in substantial costs to society [81]. This pain also affects individuals' quality of life [47] which is very closely linked to physical functioning. Therefore, physical measures of physical functioning is important to predict outcomes for neck and thoracic populations. The use of physical measures of physical functioning is increasing [36] and these measures are also used in clinical practice guidelines [21, 82].

   However, the systematic review on physical measures of physical functioning as candidate prognostic factors for predicting outcomes for neck and thoracic pain is insufficient. Predicting outcomes is crucial for both physicians and patients. Prognostic factor research is a fundamental step in developing accurate prognostic models for that purpose [83]. It is expected that the results of the systematic review would have several benefits. First, this review will address important gaps in the literature by providing a comprehensive overview of physical prognostic factors. Second, by including both conditions in this review, it will allow for a more comprehensive analysis of common and distinct prognostic factors, ultimately offering a more comprehensive understanding of the prognostic factors that impact outcomes in individuals with neck and thoracic pain. Third, this prognostic review can enable early identification of patients at risk for poor outcomes and it can play a vital role in policy and clinical decision-making [84] to improve overall quality of patient care. For example, knowledge of prognostic factors can aid in the implementation of personalized treatment for individuals with neck and thoracic pain who are less likely to respond to any specific treatment [85] that can enhance mobility and functionality, enabling patients to perform daily activities with greater ease, leading to a more active and fulfilling life. Fourth, this systematic review will also provide the best evidence for clinicians and researchers to identify the best physical outcome measures of physical functioning as prognostic factors for neck and thoracic populations. Consequently, it will lower the healthcare cost for patients [86]. Furthermore, providing better understanding of prognostic factors, patients would be aware about their condition which can help them to manage their recovery expectations [53]. These benefits can lead to better prediction models and more personalized treatment plans for patients suffering from neck and thoracic pain that can lead to faster recovery, better pain management, and an overall improvement in health-related quality of life [87]. However, the certainty of the evidence of this systematic review may be limited by the limited number of studies available and the possible low quality of the individual studies.

## Supporting information

**S1 File. PRISMA-P 2015 checklist.** This file shows the completed PRISMA-P 2015 checklist. (DOCX)

**S2 File. MEDLINE (Ovid) search strategy.** This file shows MEDLINE search strategy. (DOCX)

## Acknowledgments

Librarian, Western Libraries, Western University, London, Ontario, Canada.

Participants of the Patient Partner Advisory Group (PPAG), Spinal Pain Research in the School of Physical Therapy at Western University.

## Author Contributions

**Conceptualization:** Rabea Begum, Alison Rushton, Alaa El Chamaa, David Walton, Paul Parikh.

**Methodology:** Rabea Begum, Alison Rushton, Alaa El Chamaa, David Walton, Paul Parikh.

**Supervision:** Alison Rushton, David Walton, Paul Parikh.

**Validation:** Alison Rushton, David Walton, Paul Parikh.

**Writing – original draft:** Rabea Begum.

**Writing – review & editing:** Rabea Begum, Alison Rushton, Alaa El Chamaa, David Walton, Paul Parikh.

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
