## [Decision Letter · Decision Letter 0]

29 Sep 2024

PONE-D-24-34920Physical measures of physical functioning as prognostic factors in predicting outcomes for neck and thoracic pain: protocol for a systematic reviewPLOS ONE

Dear Dr. Begum, Thank you for submitting your manuscript to PLOS ONE. After careful consideration, we feel that it has merit but does not fully meet PLOS ONE’s publication criteria as it currently stands. Therefore, we invite you to submit a revised version of the manuscript that addresses the points raised during the review process.

We look forward to receiving your revised manuscript.

Kind regards,

Cho-Hao Howard Lee, M.D.

Academic Editor

PLOS ONE

Journal Requirements: When submitting your revision, we need you to address these additional requirements. 1. Please ensure that your manuscript meets PLOS ONE's style requirements, including those for file naming. The PLOS ONE style templates can be found at https://journals.plos.org/plosone/s/file?id=wjVg/PLOSOne_formatting_sample_main_body.pdf and https://journals.plos.org/plosone/s/file?id=ba62/PLOSOne_formatting_sample_title_authors_affiliations.pdf 2. Please include a separate caption for each figure in your manuscript. 3. Please include your tables as part of your main manuscript and remove the individual files. Please note that supplementary tables (should remain/ be uploaded) as separate ""supporting information"" files 4. Please include captions for your Supporting Information files at the end of your manuscript, and update any in-text citations to match accordingly. Please see our Supporting Information guidelines for more information: http://journals.plos.org/plosone/s/supporting-information. 5. Please review your reference list to ensure that it is complete and correct. If you have cited papers that have been retracted, please include the rationale for doing so in the manuscript text, or remove these references and replace them with relevant current references. Any changes to the reference list should be mentioned in the rebuttal letter that accompanies your revised manuscript. If you need to cite a retracted article, indicate the article’s retracted status in the References list and also include a citation and full reference for the retraction notice. 6. As required by our policy on Data Availability, please ensure your manuscript or supplementary information includes the following:  A numbered table of all studies identified in the literature search, including those that were excluded from the analyses.   For every excluded study, the table should list the reason(s) for exclusion.   If any of the included studies are unpublished, include a link (URL) to the primary source or detailed information about how the content can be accessed.  A table of all data extracted from the primary research sources for the systematic review and/or meta-analysis. The table must include the following information for each study:  Name of data extractors and date of data extraction  Confirmation that the study was eligible to be included in the review.   All data extracted from each study for the reported systematic review and/or meta-analysis that would be needed to replicate your analyses.  If data or supporting information were obtained from another source (e.g. correspondence with the author of the original research article), please provide the source of data and dates on which the data/information were obtained by your research group.  If applicable for your analysis, a table showing the completed risk of bias and quality/certainty assessments for each study or outcome.  Please ensure this is provided for each domain or parameter assessed. For example, if you used the Cochrane risk-of-bias tool for randomized trials, provide answers to each of the signalling questions for each study. If you used GRADE to assess certainty of evidence, provide judgements about each of the quality of evidence factor. This should be provided for each outcome.   An explanation of how missing data were handled.  This information can be included in the main text, supplementary information, or relevant data repository. Please note that providing these underlying data is a requirement for publication in this journal, and if these data are not provided your manuscript might be rejected. 

Reviewers' comments:

Reviewer's Responses to Questions

**Comments to the Author**

1. Does the manuscript provide a valid rationale for the proposed study, with clearly identified and justified research questions?

Reviewer #1: Partly

Reviewer #2: Yes

Reviewer #3: Yes

Reviewer #4: Partly

2. Is the protocol technically sound and planned in a manner that will lead to a meaningful outcome and allow testing the stated hypotheses?

Reviewer #1: Partly

Reviewer #2: Yes

Reviewer #3: Yes

Reviewer #4: Yes

3. Is the methodology feasible and described in sufficient detail to allow the work to be replicable?

Reviewer #1: Yes

Reviewer #2: Yes

Reviewer #3: Yes

Reviewer #4: Yes

4. Have the authors described where all data underlying the findings will be made available when the study is complete?

Reviewer #1: No

Reviewer #2: Yes

Reviewer #3: Yes

Reviewer #4: Yes

5. Is the manuscript presented in an intelligible fashion and written in standard English?

Reviewer #1: No

Reviewer #2: Yes

Reviewer #3: Yes

Reviewer #4: Yes

6. Review Comments to the Author

You may also provide optional suggestions and comments to authors that they might find helpful in planning their study.

Reviewer #1: Decision: Major Revision

Thank you for submitting this protocol on a very relevant and timely topic. The systematic review you propose has the potential to fill significant gaps in the literature regarding the role of physical measures as prognostic factors for neck and thoracic pain. While the protocol demonstrates thoughtful planning, there are a few key areas that require more substantial clarification and revision to ensure the review's rigor and applicability.

Merits:

Important Research Focus: Addressing the prognostic value of physical measures for neck and thoracic pain is both necessary and valuable. These conditions have a significant impact on patients' quality of life, and better understanding the predictive factors for outcomes could greatly improve patient care and treatment planning.

Comprehensive Search Strategy: I appreciate the extensive search plan that includes multiple databases and the inclusion of non-English studies, which helps to minimize bias. This is a major strength of the review and will ensure that a broad spectrum of evidence is captured.

Patient Involvement: Including input from the Patient Partner Advisory Group is a commendable effort to incorporate patient perspectives, ensuring the relevance and applicability of the findings to real-world practice.

Aspects Requiring Major Revision:

Inconsistencies in Terminology and Conceptual Framework: There is a lack of clarity in how you define and differentiate between the types of physical measures you will analyze. Terms like "impairment-based," "performance-based," and "activity measures in a natural environment" are used, but the definitions and distinctions between these categories need further elaboration. Additionally, how these measures will be interpreted or weighted during the synthesis of the findings remains vague. Providing a clear conceptual framework for how these different measures will be assessed and categorized is critical for readers to fully understand your approach.

Ambiguity Regarding Outcome Measures: Although the protocol outlines that "no limitation will be placed on the types of outcomes evaluated," this may lead to an overly broad and unfocused review. Consider narrowing the scope to focus on specific, clinically relevant outcomes (e.g., pain, functional mobility, quality of life) that align with the prognostic factors under investigation. This will help ensure that your results are applicable to clinical practice and not overly diluted by including less relevant outcomes.

Lack of Specificity in Data Synthesis and Meta-Analysis Plans: The manuscript would benefit from a much more detailed explanation of how you plan to handle heterogeneity, particularly statistical heterogeneity. While the I² and Chi² tests are mentioned, there’s no clear plan for how different types of heterogeneity will be managed in practice. Will you conduct subgroup analyses? If so, based on which variables? Providing a more structured approach to data synthesis and interpretation is essential, particularly for a systematic review protocol. A clearer decision tree for when you will use a quantitative meta-analysis versus a narrative synthesis would also strengthen the protocol.

Handling of Bias and Quality Assessment: While QUIPS and GRADE are mentioned for assessing the risk of bias and evidence quality, the protocol doesn’t adequately explain how these tools will be applied in practice. Will you exclude studies with high risk of bias, or will they be weighted differently in the analysis? Further detail on how these assessments will impact the inclusion or exclusion of studies is necessary. Additionally, it would be useful to specify any plans for sensitivity analysis to assess the impact of studies with high risk of bias.

Translation and Use of Automated Tools: The use of ChatGPT for translating non-English studies is innovative but comes with risks regarding the accuracy and nuance of translations. While the manuscript mentions human verification of translations, it would benefit from a more detailed explanation of how you plan to ensure that essential details, such as medical terminology, are correctly interpreted. This is particularly important given the technical nature of the studies you are including.

Ethics and Patient Involvement: While the manuscript includes a section on patient involvement, it does not sufficiently elaborate on how patients contributed to the design or methodology of the study. A more detailed account of how patient perspectives shaped the review’s focus or design, beyond simply validating it, would add depth to the protocol and ensure that it aligns with patient-centered research principles.

Conclusion:

This manuscript has significant potential but requires substantial revision before it can move forward. The key areas that need attention include clarifying your conceptual framework and definitions of physical measures, providing more detail on how you will handle heterogeneity and bias, and refining the scope of outcome measures to ensure a focused and clinically relevant review. Additionally, the use of automated tools for translation requires careful oversight to avoid inaccuracies. With these major revisions, the manuscript will be much better positioned to provide robust and actionable insights for clinicians and researchers.

I recommend major revisions to address the concerns outlined above.

Thank you for your work, and I look forward to reviewing the revised manuscript.

Reviewer #2: This protocol outlines a systematic review aimed at summarizing evidence for physical measures of physical functioning as prognostic factors for outcomes in individuals with neck and thoracic pain. The authors propose a comprehensive search strategy across multiple databases, including grey literature, to identify prospective longitudinal cohort studies. They plan to assess risk of bias using the QUIPS tool and evaluate the quality of evidence using the GRADE approach. The review aims to address an important gap in the literature by providing a comprehensive overview of physical prognostic factors for neck and thoracic pain, which could inform clinical decision-making and patient care.

Limited discussion on handling of missing data: The protocol does not adequately address how missing data will be handled, which is crucial in prognostic studies. This omission could affect the quality and reliability of the review's findings. For example: If studies report standard errors without reporting standard deviations, will you attempt to estimate the missing standard deviations using appropriate statistical tools and calculators?

Despite this limitation, this systematic review protocol addresses an important gap in the literature and has the potential to provide valuable insights into physical prognostic factors for neck and thoracic pain outcomes.

Reviewer #3: This manuscript presents a well-structured protocol for a systematic review aimed at summarizing evidence regarding physical measures of physical functioning as prognostic factors in individuals with neck and thoracic pain. The topic is relevant, as neck and thoracic pain are significant public health concerns, and identifying prognostic factors could help improve patient outcomes and clinical decision-making. The methodology appears rigorous, and the authors have followed recognized guidelines such as PRISMA-P, which strengthens the validity of their approach. However, there are a few areas need improvements.

Major Points:

1. The manuscript does an excellent job highlighting the gap in the literature, particularly the need for a review that covers both neck and thoracic pain. However, the rationale for the inclusion of thoracic pain in this review could be further elaborated. Since thoracic pain is less studied compared to neck pain, an explicit discussion about how insights from thoracic pain will be combined with those from neck pain would strengthen the introduction. Will these two conditions be analyzed separately or together?

2. The manuscript mentions categorizing outcomes into impairment-based, performance-based, and activity measures. While this is an appropriate approach, more information is needed on how the authors will handle studies that assess these measures using different instruments. Will they attempt to standardize or pool data from different measurement tools, and if so, how? Additionally, it would be useful to include examples of specific outcome measures under each category to give readers a clearer sense of what is included.

3. The criteria for excluding studies based on non-neuromusculoskeletal origins of pain are appropriate, but the exclusion criteria regarding metabolic bone diseases such as osteoporosis could be reconsidered. Given that osteoporosis is common in older adults, and that neck and thoracic pain are also prevalent in this population, the exclusion may inadvertently omit relevant studies. The authors should either justify this decision more explicitly or consider including such studies with careful adjustment for confounding variables.

4. The authors have selected appropriate tools (QUIPS, GRADE) for assessing the risk of bias and quality of evidence. It would be beneficial to describe in more detail how the authors plan to integrate the findings from the QUIPS tool into the overall analysis. For example, will high-risk studies be excluded from the meta-analysis, or will they be included with sensitivity analyses?

Reviewer #4: The manuscript outlines a protocol for a systematic review investigating the use of physical measures of physical functioning as prognostic factors in predicting outcomes in individuals with neck and thoracic pain. The protocol is well-structured, and the methodology is clearly defined and aligned with best practices for systematic reviews. However, there are areas that require clarification and improvement to strengthen the protocol and ensure robust results.

1. The QUIPS tool is appropriately selected for risk of bias assessment. However, the manuscript should clarify how discrepancies in bias assessments will be handled between the two reviewers beyond consulting a third reviewer. A more structured approach, such as predefined criteria for resolving differences in bias ratings, could improve consistency and reduce potential reviewer bias.

2. The protocol plans to include studies in multiple languages and use tools like ChatGPT for translation. While this is commendable, it is important to ensure the accuracy of translations in technical and clinical terminology. It is recommended to provide more details on the verification process for translated studies, especially how bilingual reviewers will be selected and how translation accuracy will be validated.

3. While the manuscript states that a meta-analysis will be performed if studies show homogeneity, it would benefit from a clearer explanation of how the authors will determine whether studies are sufficiently homogenous. For instance, more information on the thresholds for heterogeneity that will guide decisions to pool data would be useful. Additionally, the narrative synthesis section should be expanded to clarify how qualitative data will be synthesized, categorized, and presented if a meta-analysis is not possible.

4. The manuscript states that authors will be contacted for missing or unclear data, but it does not sufficiently explain how missing data will be handled if authors do not respond. A more detailed description of the imputation methods or sensitivity analyses for handling missing data would strengthen the robustness of the review.

5. While the protocol defines short-, medium-, and long-term follow-up periods, it does not specify how these timeframes will be standardized across different studies that may use varying definitions for follow-up. Including a more detailed strategy for dealing with differing follow-up periods, or defining a minimum acceptable follow-up length, would enhance comparability across studies.

7. PLOS authors have the option to publish the peer review history of their article (what does this mean?). If published, this will include your full peer review and any attached files.

Reviewer #1: **Yes: **Xiaoyi Zhang

Reviewer #2: No

Reviewer #3: No

Reviewer #4: **Yes: **Yuhang Liu

---

## [Author Response · Author response to Decision Letter 0]

10 Nov 2024

Rebuttal Letter

To,

The Editor, Reviewers,

PLOS ONE

Date: November 10, 2024

Subject: Addressing Editor and Reviewer Comments for PONE-D-24-34920.

Dear Editor and Reviewers,

We are pleased to have an opportunity to revise our manuscript titled “Physical measures of physical functioning as prognostic factors in predicting outcomes for neck and thoracic pain: protocol for a systematic review”. In our revised manuscript, we have carefully considered the editor’s and reviewers’ suggestions, to address all your concerns to the best of our abilities. We provide our responses to the suggestions below. Comments from Editor and Reviewers are colored in red, and our responses are highlighted in black. We have also updated our S1 file that shows the PRISMA-P checklist.

All feedback on our manuscript were very helpful, and we greatly appreciate your constructive feedback on our original submission. After addressing the issues raised, we feel the quality of the manuscript is much improved.

Sincerely,

Rabea Begum

On behalf of authors of PONE-D-24-34920

Response to Editor

• A rebuttal letter that responds to each point raised by the academic editor and reviewer(s). You should upload this letter as a separate file labeled ‘Response to Reviewers’.

o Included

• A marked-up copy of your manuscript that highlights changes made to the original version. You should upload this as a separate file labeled ‘Revised Manuscript with Track Changes’.

o Included

• An unmarked version of your revised paper without tracked changes. You should upload this as a separate file labeled ‘Manuscript’.

o Included

Journal Requirements:

1. Please ensure that your manuscript meets PLOS ONE’s style requirements, including those for file naming. The PLOS ONE style templates can be found at 

 Action(s) performed: We made changes to the new version of the manuscript according to the submission guidelines and PLOS ONE’s style requirements (lines 13,14) including those for file naming. We would like to inform you that we also used the tab key for paragraph indentation during the editing process.

2. Please include a separate caption for each figure in your manuscript.

Action(s) performed: We formatted in-text figure caption as “Fig 1” in our manuscript (line 221) according to PLOS ONE’s requirements in. We also included a figure caption with legend as “Fig 1. PRISMA 2020 flow diagram. This diagram reflects the study selection process” (line 222). In addition, we changed the separate Fig 1 file as “Fig 1.tiff”. 

3. Please include your tables as part of your main manuscript and remove the individual files. Please note that supplementary tables (should remain/ be uploaded) as separate “supporting information” files

Action(s) performed: We removed the individual file, and the manuscript was adjusted accordingly with adding Table 1 (lines 234, 235).

 Action(s) performed: We added supporting information files at the end of our revised manuscript (lines 657 to 660). We updated our revised manuscript with an in-text S1 file (line 150). We also renamed separate supporting information files as S1_file.docx and S2_file.docx.

Action(s) performed: We have updated our reference list. 

6. As required by our policy on Data Availability, please ensure your manuscript or supplementary information includes the following: 

 If applicable for your analysis, a table showing the completed risk of bias and quality/certainty assessments for each study or outcome. Please ensure this is provided for each domain or parameter assessed. For example, if you used the Cochrane risk-of-bias tool for randomized trials, provide answers to each of the signaling questions for each study. If you used GRADE to assess certainty of evidence, provide judgements about each of the quality of evidence factor. This should be provided for each outcome. 

Action(s) performed: None as the above points are not relevant to a protocol manuscript except dealing with missing data. We updated our manuscript (lines 317-325).

Responses to Reviewer 1

Inconsistencies in Terminology and Conceptual Framework: There is a lack of clarity in how you define and differentiate between the types of physical measures you will analyze. Terms like "impairment-based," "performance-based," and "activity measures in a natural environment" are used, but the definitions and distinctions between these categories need further elaboration. Additionally, how these measures will be interpreted or weighted during the synthesis of the findings remains vague. Providing a clear conceptual framework for how these different measures will be assessed and categorized is critical for readers to fully understand your approach.

Action(s) performed: Thank you for your feedback regarding the clarity of terminology and the conceptual framework in our review. We appreciate your emphasis on the need for well-defined categories. We have elaborated on the categories of physical measures, including impairment-based, performance-based, and activity measures in a natural environment, ensuring comprehensive coverage in the protocol. We have revised the relevant paragraph (lines 77 to 96). 

Ambiguity Regarding Outcome Measures: Although the protocol outlines that "no limitation will be placed on the types of outcomes evaluated," this may lead to an overly broad and unfocused review. Consider narrowing the scope to focus on specific, clinically relevant outcomes (e.g., pain, functional mobility, quality of life) that align with the prognostic factors under investigation. This will help ensure that your results are applicable to clinical practice and not overly diluted by including less relevant outcomes.

Action(s) performed: We appreciate your insightful feedback regarding the outcome measures in our systematic review protocol. We acknowledge the concern about potential ambiguity and dilution of focus due to the inclusion of a broad range of outcomes. However, we believe that including all types of outcomes is essential for our review. From our scoping searches exploring the potential value of this review, we know that the number of studies on neck pain is relatively low compared to other musculoskeletal conditions (e.g., low back pain). There is a significant gap in literature concerning neck and thoracic pain and its prognostic factors. The scarcity of studies limits our understanding of how various factors influence outcomes in this population. Owing to the low number of studies exist, a broad scope in our review was necessary and one way in which this was enabled was through no limitation on outcomes. We have provided further rationale for selecting a broader range of outcomes (lines 171, 172).

Lack of Specificity in Data Synthesis and Meta-Analysis Plans: The manuscript would benefit from a much more detailed explanation of how you plan to handle heterogeneity, particularly statistical heterogeneity. While the I² and Chi² tests are mentioned, there’s no clear plan for how different types of heterogeneity will be managed in practice. Will you conduct subgroup analyses? If so, based on which variables? Providing a more structured approach to data synthesis and interpretation is essential, particularly for a systematic review protocol. A clearer decision tree for when you will use a quantitative meta-analysis versus a narrative synthesis would also strengthen the protocol.

Action(s) performed: We appreciate your thoughtful criticism of our Data Synthesis and Meta-Analysis Plans. In our manuscript, we have mentioned a comprehensive and structured approach to data synthesis that incorporates the potential for both quantitative and qualitative analyses. We have further defined the assessment of heterogeneity. Based on our initial scoping search the subgroup analysis might not be appropriate. We updated the data synthesis part (lines 271 to 325).

Handling of Bias and Quality Assessment: While QUIPS and GRADE are mentioned for assessing the risk of bias and evidence quality, the protocol doesn’t adequately explain how these tools will be applied in practice. Will you exclude studies with high risk of bias, or will they be weighted differently in the analysis? Further detail on how these assessments will impact the inclusion or exclusion of studies is necessary. Additionally, it would be useful to specify any plans for sensitivity analysis to assess the impact of studies with high risk of bias.

Action(s) performed: We appreciate your valuable feedback. We understand your concern regarding risk of bias and quality assessment. We have elaborated on the use of the QUIPS tool for assessing the risk of bias in individual studies based on your suggestion (lines 243 to 264). Studies with a high RoB will not be excluded solely based on their risk rating; however, their influence will be carefully considered. Studies rated as having a high RoB in critical domains may be given less weight in the synthesis of evidence, ensuring that the final conclusions are based on more robust findings. We will also assess whether studies with a higher RoB contribute to downgrading the overall quality of evidence during the GRADE assessment. We might not perform sensitivity analysis due to limited number of studies identified in our scoping searches. We have updated our manuscript (295 to 300).

Translation and Use of Automated Tools: The use of ChatGPT for translating non-English studies is innovative but comes with risks regarding the accuracy and nuance of translations. While the manuscript mentions human verification of translations, it would benefit from a more detailed explanation of how you plan to ensure that essential details, such as medical terminology, are correctly interpreted. This is particularly important given the technical nature of the studies you are including.

Action(s) performed: Thank you for raising the important concern regarding the use of ChatGPT for translating non-English studies, especially considering the technical nature of the studies included in our review. We fully acknowledge the limitations and potential risks associated with automated translation tools, particularly in the context of medical and scientific terminology. To address this, after the automated translation, bilingual experts with proficiency in both the source language and English will carefully review the translated material. These individuals will have a background in medical or health sciences, allowing them to identify and correct any errors or ambiguities, particularly with technical or field-specific terminology. We have elaborated in detail in the study design section (lines 180, 181).

Ethics and Patient Involvement: While the manuscript includes a section on patient involvement, it does not sufficiently elaborate on how patients contributed to the design or methodology of the study. A more detailed account of how patient perspectives shaped the review’s focus or design, beyond simply validating it, would add depth to the protocol and ensure that it aligns with patient-centered research principles.

Action(s) performed: Thank you for your valuable feedback. We acknowledge the importance of patient involvement in ensuring the study aligns with patient-centered research principles. In response to your comment, we would like to clarify that the advisory groups expressed their lived experiences with neck and thoracic pain, and they highlighted the importance of this protocol which ensured that it would be understandable and usable for both researchers and patients. Their feedback also guided the selection and inclusion of outcomes that reflect a broader range of experiences with spinal pain. Specifically, their feedback highlighted the need to include not only traditional clinical outcomes (e.g., pain, disability) but also those that reflect day-to-day functioning and quality of life, which are highly relevant to individuals living with neck and thoracic pain. This is another justification we have made efforts to include a diverse set of outcomes. While they found no need for modifications to the protocol, their feedback reassured us that the methodology and design resonated with their experiences. We adjusted our manuscript (lines 368 to 377).

Responses to Reviewer 2

Limited discussion on handling of missing data: The protocol does not adequately address how missing data will be handled, which is crucial in prognostic studies. This omission could affect the quality and reliability of the review's findings. For example: If studies report standard errors without reporting standard deviations, will you attempt to estimate the missing standard deviations using appropriate statistical tools and calculators?

Action(s) performed: Thank you for your valuable feedback regarding the handling of missing data in our systematic review protocol. We have added a separate paragraph on how we will handle missing data (lines 317 to 325). By implementing this process, we aim to minimize the impact of missing data on our review. We believe this method ensures thoroughness and transparency, allowing us to clearly report on the extent of missing data and its implications for our conclusions.

Responses to Reviewer 3

Major Points:

1. The manuscript does an excellent job highlighting the gap in the literature, particularly the need for a review that covers both neck and thoracic pain. However, the rationale for the inclusion of thoracic pain in this review could be further elaborated. Since thoracic pain is less studied compared to neck pain, an explicit discussion about how insights from thoracic pain will be combined with those from neck pain would strengthen the introduction. Will these two conditions be analyzed separately or together?

Ac

---

## [Decision Letter · Decision Letter 1]

18 Dec 2024

Physical measures of physical functioning as prognostic factors in predicting outcomes for neck and thoracic pain: protocol for a systematic review

PONE-D-24-34920R1

Dear Dr. Rabea Begum,

We’re pleased to inform you that your manuscript has been judged scientifically suitable for publication and will be formally accepted for publication once it meets all outstanding technical requirements.

Kind regards,

Cho-Hao Howard Lee, M.D.

Academic Editor

PLOS ONE

Reviewers' comments:

Reviewer's Responses to Questions

**Comments to the Author**

1. Does the manuscript provide a valid rationale for the proposed study, with clearly identified and justified research questions?

Reviewer #1: Partly

Reviewer #2: Yes

Reviewer #3: Yes

2. Is the protocol technically sound and planned in a manner that will lead to a meaningful outcome and allow testing the stated hypotheses?

Reviewer #1: Partly

Reviewer #2: Yes

Reviewer #3: Yes

3. Is the methodology feasible and described in sufficient detail to allow the work to be replicable?

Reviewer #1: Yes

Reviewer #2: Yes

Reviewer #3: Yes

4. Have the authors described where all data underlying the findings will be made available when the study is complete?

Reviewer #1: Yes

Reviewer #2: Yes

Reviewer #3: Yes

5. Is the manuscript presented in an intelligible fashion and written in standard English?

Reviewer #1: Yes

Reviewer #2: Yes

Reviewer #3: Yes

6. Review Comments to the Author

You may also provide optional suggestions and comments to authors that they might find helpful in planning their study.

Reviewer #1: Merits

1. Relevance and Timeliness: This protocol addresses a critical need for systematic evidence regarding physical functioning measures, providing insights applicable to a wide range of musculoskeletal conditions.

2. Rigorous Methodology: The adherence to Cochrane guidelines and the PRISMA-P framework ensures transparency and reliability, making this protocol a strong foundation for the planned systematic review.

3. Practical Implications: By examining prognostic factors for neck and thoracic pain, the study has the potential to inform clinical guidelines and improve the management of a widespread and burdensome condition.

Minor Suggestions for Revision

1. Clarify Conceptual Framework for Physical Measures: The manuscript discusses impairment-based, performance-based, and activity measures but could benefit from a more explicit conceptual framework. A schematic or brief table summarizing these categories, including examples and their relevance to outcomes, would aid reader understanding.

2. Expand on Clinical Implications: While the importance of understanding prognostic factors is noted, the discussion could be enhanced by outlining specific ways clinicians might use this information in practice (e.g., tailoring rehabilitation programs based on predictive physical measures).

3. Strengthen the Patient Involvement Section: The inclusion of patient advisory feedback is commendable. Providing more detail on how this input shaped the design, particularly regarding outcome selection and applicability, would align with patient-centered research principles.

4. Refine the Approach to Missing Data: The plan for handling missing data is sound but could be made more robust by outlining contingency strategies if significant data remain unavailable (e.g., exclusion of studies with critical gaps).

Recommendation: Minor Revision

This manuscript is well-prepared and addresses an important topic in musculoskeletal research. The suggested revisions aim to clarify certain aspects and enhance the overall impact of the protocol. Thank you for your thoughtful work, and I look forward to seeing the results of this review!

Reviewer #2: this systematic review protocol addresses an important gap in the literature and has the potential to provide valuable insights into physical prognostic factors for neck and thoracic pain outcomes. I don't have extra comments.

Reviewer #3: The authors have adequately addressed all of my concerns. I support the publication of this manuscript. Best of luck!

7. PLOS authors have the option to publish the peer review history of their article (what does this mean?). If published, this will include your full peer review and any attached files.

Reviewer #1: **Yes: **Xiaoyi Zhang

Reviewer #2: No

Reviewer #3: No

---

## [Editor Report · Acceptance letter]

10 Jan 2025

PONE-D-24-34920R1 

PLOS ONE

Dear Dr. Begum, 

I'm pleased to inform you that your manuscript has been deemed suitable for publication in PLOS ONE. Congratulations! Your manuscript is now being handed over to our production team.

Kind regards, 

on behalf of

Dr. Cho-Hao Howard Lee 

Academic Editor

PLOS ONE